# Big-Volume SliceGAN for Improving a Synthetic 3D Microstructure Image of Additive-Manufactured TYPE 316L Steel

**DOI:** 10.3390/jimaging9050090

**Published:** 2023-04-29

**Authors:** Keiya Sugiura, Toshio Ogawa, Yoshitaka Adachi, Fei Sun, Asuka Suzuki, Akinori Yamanaka, Nobuo Nakada, Takuya Ishimoto, Takayoshi Nakano, Yuichiro Koizumi

**Affiliations:** 1Department of Material Design Innovation Engineering, Nagoya University, Nagoya 464-8603, Japan; 2Division of Mechanical Systems Engineering, Tokyo University of Agriculture and Technology, Tokyo 184-8588, Japan; 3School of Materials and Chemical Technology, Tokyo Institute of Technology, Tokyo 226-8503, Japan; 4Department of Materials Design and Engineering, Toyama University, Toyama 930-8555, Japan; 5Division of Materials and Manufacturing Science, Osaka University, Osaka 565-0871, Japan

**Keywords:** SliceGAN, generative adversarial network, synthetic 3D image, additive manufacturing, autocorrelation function

## Abstract

A modified SliceGAN architecture was proposed to generate a high-quality synthetic three-dimensional (3D) microstructure image of TYPE 316L material manufactured through additive methods. The quality of the resulting 3D image was evaluated using an auto-correlation function, and it was discovered that maintaining a high resolution while doubling the training image size was crucial in creating a more realistic synthetic 3D image. To meet this requirement, modified 3D image generator and critic architecture was developed within the SliceGAN framework.

## 1. Introduction

The reconstruction of three-dimensional (3D) microstructures can enhance our understanding of the properties of a material. Traditionally, serial sectioning [1,2] and tomography [3] have been used to generate 3D microstructure images. However, these methods are time-consuming and require specialized equipment. Recently, Kench and Cooper [4] introduced a new approach for efficient 3D microstructure reconstruction using a generative adversarial network (GAN) called SliceGAN. There are two primary types of image generation algorithms: adversarial generation network (GAN) [5] and variational autoencoder [6]. SliceGAN produces a synthetic 3D image from one or three two-dimensional (2D) images for isotropic and anisotropic microstructures, respectively.

SliceGAN consists of three components: a 3D image generator (3D generator), a critic (similar to a discriminator in conventional GAN [5]), and a slicer. The 3D generator creates a 3D image from noise (latent variables), which is then sliced into three perpendicular planes by the slicer. The critic compares the sliced images with 2D images cropped from an original microstructure image, and updates the weight coefficient in the transpose convolution matrix of the 3D generator accordingly. SliceGAN runs on a high-performance graphics processing unit in Pytorch frame and GPGPU mode.

In the original SliceGAN architecture proposed by Kench and Cooper [4], 64 sets of latent variables in the format of 4 × 4 × 4 (voxel) were used. These latent variables were processed by a transpose convolution with five layers, resulting in a 3D image with dimensions of 64 × 64 × 64 voxels and three channels. The 2D images sliced from the generated 3D image were compared with 2D images cropped from the original image using the critic of Wasserstein GAN with Gradient Penalty (WGAN-GP) [7]. The weight coefficient of the 3D generator was then updated based on the result. The high performance of SliceGAN is partially due to the architecture of WGAN-GP. This process is repeated until a given epoch number is reached. The resolution of a 3D image generated by the default SliceGAN is sufficient for a binary image [8], but inadequate for a grayscale image. Since actual microstructural photographs are grayscale images rather than binary images, it would be more useful in material research to improve the quality of a SliceGAN-generated image using grayscale 2D images. In particular, generating a large representative volume while maintaining high resolution is beneficial for reproducing a representative microstructure.

The quality of a synthetic 3D microstructure image generated by SliceGAN may suffer from image degradation caused by the small cropped image size (64 × 64 pixels) and/or limited layers of the transpose convolution. Therefore, it is necessary to consider enlarging the cropped image size and/or modifying the transpose convolution architecture. In addition, the quality of a 3D image generated by SliceGAN should be evaluated quantitatively to achieve better results.

To evaluate the performance of SliceGAN, the quality of the generated 3D image should be assessed quantitatively. However, there is currently little established method for evaluating grayscale 3D images. Kench and Cooper [4] suggested a possible approach to measure the similarity between an experimentally obtained 3D image and a synthetic 3D image generated by SliceGAN. They used tomographic 3D data collected from a Li-ion MMC cathode sample and trained SliceGAN with a random subset of 2D sections of the 3D image. They then evaluated the similarity between the real and synthetic 3D images based on volume fraction, relative surface area, and relative diffusivity. This assessment demonstrated that SliceGAN can potentially generate a 3D image that is similar to the ground truth. However, their evaluation only used binary 3D images, and the quality of synthetic 3D images with grayscale sections is still under debate.

To investigate the anisotropic microstructure of additive-manufactured TYPE 316L stainless steel, a modified architecture of SliceGAN is proposed in this study to enhance the quality of fake 3D microstructure images. Furthermore, the quality of these images is quantitatively evaluated.

## 2. Modified Architecture of SliceGAN

Table 1 and Table 2 and Figure 1 compare the architectures of the original SliceGAN (referred to as model_64) [4] and the modified SliceGAN (model_128). In the original model_64, 64 × 64 pixel images are cropped from the original 2D image and compared with 64 × 64 pixel images sliced from a generated fake 3D image by the critic. The original model uses five layers of transpose convolution to generate a 1 channel × 64 × 64 × 64 voxel 3D fake image from 64 sets of latent variables in the format of 4 × 4 × 4.

In contrast, the modified model_128 uses 128 × 128 pixel images cropped from the original 2D image and compared with 128 × 128 pixel images sliced from a generated fake 3D image by the critic. Enlarging the cropped image is beneficial because it contains a large representative volume of elements, maintaining its resolution. The modified model uses six layers of transpose convolution to generate a 1 channel × 128 × 128 × 128 pixel 3D fake image from 64 sets of latent variables in the format of 4 × 4 × 4. This larger volume SliceGAN enhances the quality of the generated 3D fake image because larger images with representative features are used in training.

Figure 2 shows the hyperparameter update process for the 3D generator and critic. In one epoch, the critic is trained five times using cropped 2D and fake sliced images, while the 3D generator is trained once. After repeated training, 64 sets of latent variables in the format of 18 × 18 × 18 and 10 × 10 × 10 voxel are input to the 3D generators of model_64 and model_128, respectively, to obtain a larger 3D synthetic image with a 512 × 512 × 512 voxel multiplied by 1 channel. Both models use thirty-two subdivided datasets (batch size) for generator training, performed only once. Meanwhile, eight subdivided datasets are used for critic training, which is repeated five times in one iteration. The optimizer used is Adam, as in the original SliceGAN model [4].

This study required the use of a high-end GPU device, the “RTX A6000,” which has 48 GB of GPU memory, to run the model_128 SliceGAN in the Pytorch framework. On the other hand, the GeForce RTX 3090 GPU with 24 GB of memory was sufficient for model_64.

## 3. 3D Synthetic Image Quality Evaluation

To assess the quality of the synthetic 3D image generated in this study, a quantitative evaluation was performed by comparing it to the original image. The microstructure under investigation was the additive-manufactured TYPE 316L steel, which exhibits a characteristic pattern known as the “crystallographic lamellar microstructure (CLM)” [9]. The periodicity of this CLM is a unique fingerprint of the microstructure because it corresponds to the laser scanning region, which has a constant interval of 80 μm. To evaluate the periodicity of the CLM, the auto-correlation function (ACF) [10] was used, which has an advantage over the fast Fourier transform (FFT) [11] in evaluating imperfect periodicity.

When an image is periodic, there are regions where the image matches well at a certain distance shift after the image is shifted from the original image. The ACF expresses the relationship between the overlap and staggered distance (called lag) of these images. The conventional correlation (*r_xy_*) is somewhat modified for ACF because X and Y are the same, except for the lag, and is given by the following equation:$$r_{xy}=\sum_{i=1}^{n}\frac{\left[X_{i}-\bar{X}\right]\left[Y_{i}-\bar{Y}\right]}{S_{x}S_{y}}$$

The equation for conventional correlation uses sample means ($\bar{X}$ and $\bar{Y}$) and sample standard deviations ($S_{x}$ and $S_{y}$) and is modified for ACF as X and Y are the same, except for the lag. To compute ACF, two images are shifted along the direction perpendicular to CLM. Another approach to compare a SliceGAN-generated image with a real image is by analyzing the brightness distribution. If the images are similar, the brightness distribution, such as the mean and standard deviation of brightness, should also be similar.

## 4. Materials

In this study, a specimen was prepared using the laser powder bed fusion (LPBF) method at Osaka University [12,13,14,15]. The LPBF process produces a unique microstructure due to the steep temperature gradient and high solidification rate. TYPE 316L powder was used, and the powder was scanned by a laser only in the X direction using Scan strategy X [12] (Figure 3). Laser power (P), scanning speed (v), and hatching distance were set at 250 W, 1000 mm/s, and 80 µm, respectively. The microstructure was highly anisotropic compared to the dual-phase microstructures of steels [8]. Therefore, a microstructure image was captured by an optical microscope on three perpendicular planes, which were used as input for training SliceGAN. It should be noted that the microstructure images used in this study were grayscale. Thus, producing a synthetic 3D microstructural image from a grayscale 2D image requires improving the spatial resolution of SliceGAN.

## 5. Discussion

In Figure 4, we can see 2D microstructure images obtained from three different sections. Figure 5 shows a pseudo-3D image created by manually combining these three sections. This pseudo-3D image is helpful for understanding the relationship between the three sections and will be compared with a 3D image generated by SliceGAN later on. In the YZ section, we observed a melt pool boundary with a scaly appearance and a thin layer in the Z-direction that corresponds to the CLM. These CLMs are located at a nearly constant pitch that corresponds to the laser hatching distance. Based on the inverse pole figure orientation map, the entire region except for CLM has approximately the same orientation.

Figure 6 presents 3D synthetic images generated by model_64 and model_128. The sections on the YZ, XY, and XZ planes of the generated 3D images are shown in Figure 7 and Figure 8. Model_128 produced a higher quality image compared to model_64, particularly for CLM, which appears more continuous in the model_128 image than in the model_64 image. Although a 128 × 128 (pixel) image was used as input for the training of SliceGAN, it is surprising that some CLMs appear continuous in the 512 × 512 (pixel) section of the model_128 image. To quantitatively evaluate the continuity of CLM observed in SliceGAN images, the length of fifty CLMs was measured for each image of model_64 and model_128 on the YZ plane. The potential maximum length of CLM is 512 (pixel) due to the image size. Figure 9 shows that the CLM in the model_128 image is much longer than that in the model_64 image. The average lengths of CLM in the image and in model_64 images are 233 and 83 (pixel), respectively. These findings suggest that enlarging cropped images to contain representative features is crucial for improving the quality of synthetic 3D images. At the same time, a 3D generator must generate a bigger synthetic 3D image by increasing transpose convolution layers. However, larger cropped images than 128 × 128 (pixel) may further improve the quality of synthetic 3D images, but much more GPU memory than 48 GB is likely required for processing the image reconstruction.

As a latent variable, a 64 (channel) × 15 × 15 random noise was input into a 3D generator.

In Figure 10, ACF results for the sectioned model_64 image, sectioned model_128 image, and the original 2D image are presented. All images were resized to 148 × 148 (pixels). The signal for model_128 appears steeper than that of model_64, and the first peak is located at the lag around 18 pixels, which is consistent with the actual interval of CLM. These ACF results suggest that model_128 can produce higher-quality fake 3D images than model_64. However, when compared to the ACF result for the original image, the ACF signal for model_128 is attenuated at larger lags. Indeed, CLM appears to be terminated in some regions, even in the sectioned model_128 image, while it is continuous in the original image. To further improve the fake 3D image, an input image larger than 128 × 128 pixels is required, as mentioned above. To achieve such a bigger-volume SliceGAN calculation, NVLink, which bridges multi-GPU devices, might be useful.

There are various characterization methods to quantify the metallurgically important characteristics of a 2D/3D image, such as grain size, area/volume fraction, particle density, connectivity, branching points, periphery, fractal dimension, and preferential direction. In addition, persistent homology has recently been used to evaluate more complicated patterns [16]. This mathematical approach is very useful in quantifying morphology in many fields. However, these methods are mainly applied to binary images. Therefore, in this study, we analyzed the density distribution of brightness values (Figure 11). The density distribution of brightness values for the three images (original, model_64, and model_128 images) is considered to be a Gaussian distribution with one peak. The density distribution of the Model_128 SliceGAN image is similar to that of the original image, while that of the Model_64 SliceGAN image appears to be wider. To examine the brightness distribution more quantitatively, we obtained the mean value and standard deviation of brightness for the three images, and the results are presented in Table 3. Both the mean value and standard deviation are quite similar between the original image and the Model_128 SliceGAN image. The mean and standard deviation of the brightness distribution of the Model_64 SliceGAN image are slightly larger than those of the original image. In the future, a more quantitative analysis of gray-scale 2D/3D images is desired.

One of RGB values is a kind of grayscale brightness value. Therefore, a high-quality synthetic color 3D image generated by SliceGAN seems to be very attractive. Kench and Cooper [4] demonstrated that SliceGAN could generate a 3D image from 2D color images as well. However, the quality of the color 3D SliceGAN image seems to be not good compared with a binary image. To improve the quality of the color 3D SliceGAN image, “Big-volume SliceGAN” is likely useful. However, it requires much more GPU memory than 48 GB and it is beyond the scope of this study.

## 6. Conclusions

This study aims to enhance the quality of synthetic 3D microstructural images generated by SliceGAN for additive-manufactured TYPE 316L steel. The effectiveness of enlarging the input image for training SliceGAN has been discovered in improving image quality. In addition to input image size, a modified 3D generator and critic architecture have also been demonstrated. The use of ACF has proven to be an effective method for quantitatively evaluating the quality of a generated 3D image.

## Figures and Tables

**Figure 1 jimaging-09-00090-f001:**
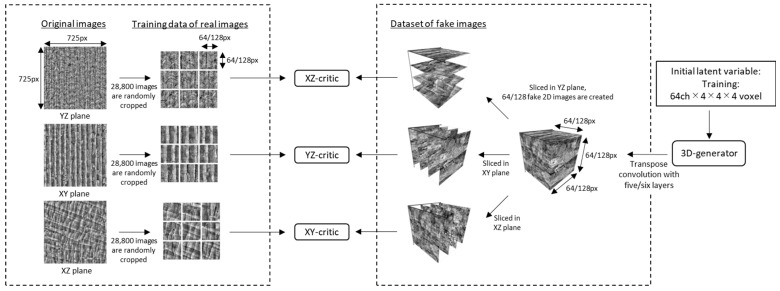
Architecture of model_64 and model_128 SliceGAN.

**Figure 2 jimaging-09-00090-f002:**
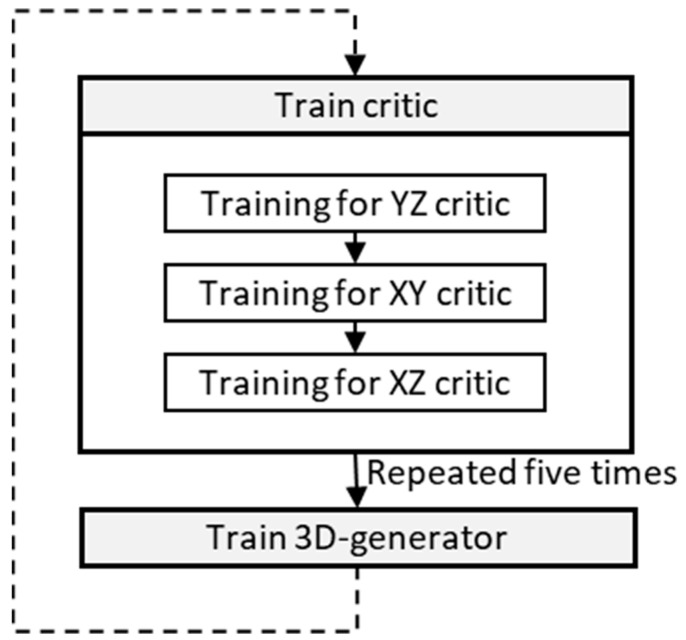
Procedure of a hyperparameter update for the 3D generator and critic.

**Figure 3 jimaging-09-00090-f003:**
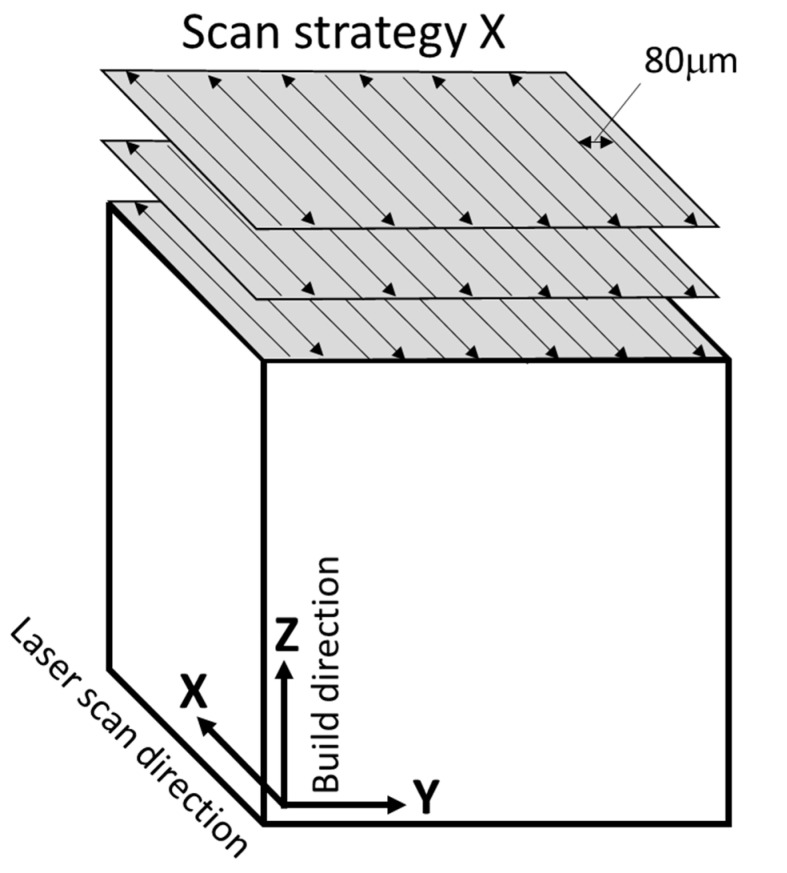
Laser scan strategy X (without rotation) [12].

**Figure 4 jimaging-09-00090-f004:**
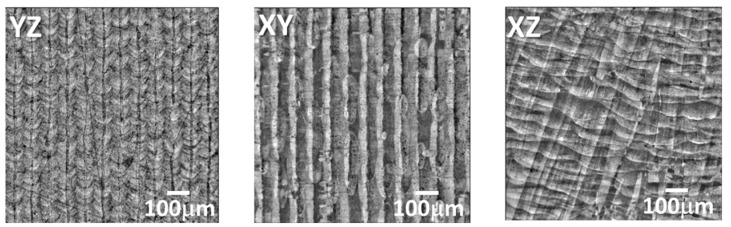
The 2D microstructures obtained from three perpendicular sections.

**Figure 5 jimaging-09-00090-f005:**
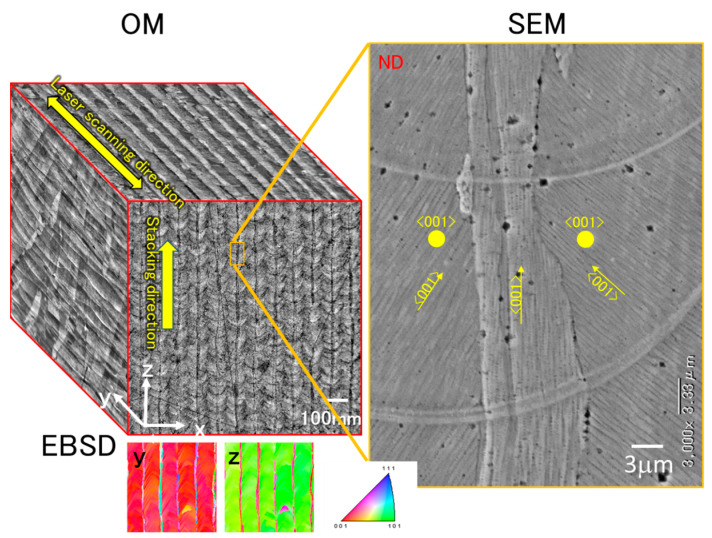
Pseudo-3D 3D optical microscope (OM) image synthesized by manually combining the three perpendicular cross-sectional images. A magnified scanning electron microscope image and inverse pole figure map are also shown.

**Figure 6 jimaging-09-00090-f006:**
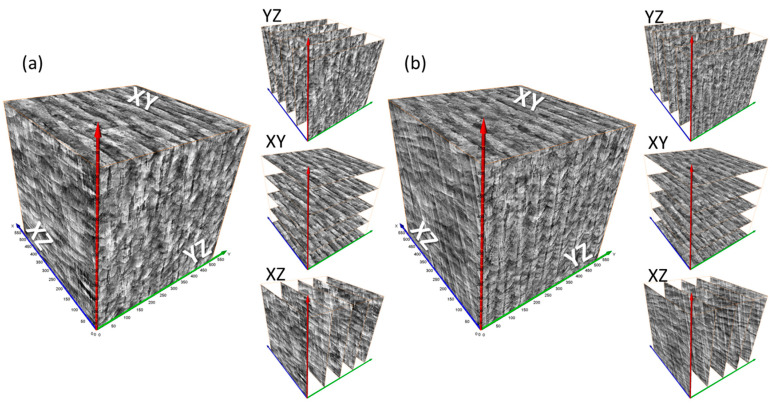
Fake 3D images generated by (**a**) model_64 and (**b**) model_128 SliceGAN.

**Figure 7 jimaging-09-00090-f007:**
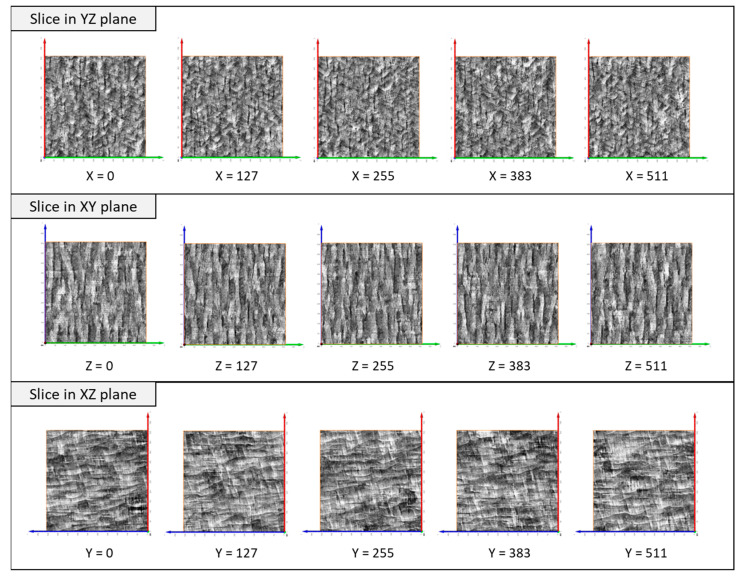
Cross-section images of a 3D image generated by model_64 SliceGAN. As a latent variable, a 64 (channel) × 18 × 18 random noise was input into a 3D generator.

**Figure 8 jimaging-09-00090-f008:**
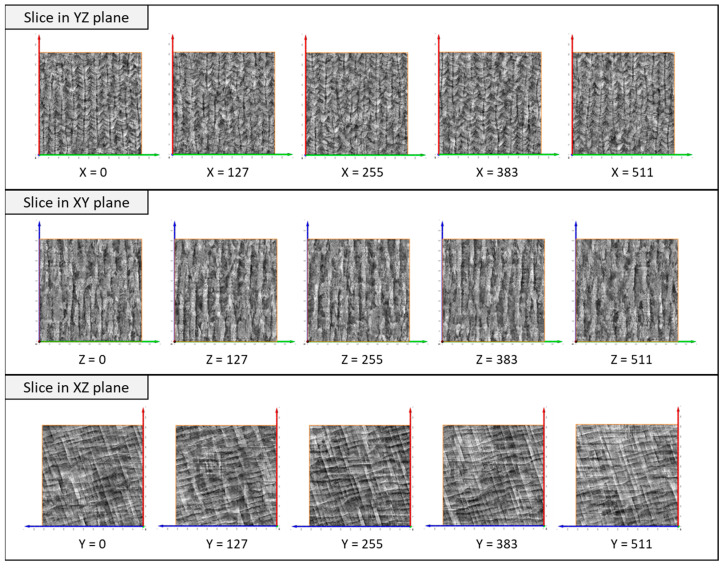
Cross-section images of a 3D image generated by model_128 SliceGAN.

**Figure 9 jimaging-09-00090-f009:**
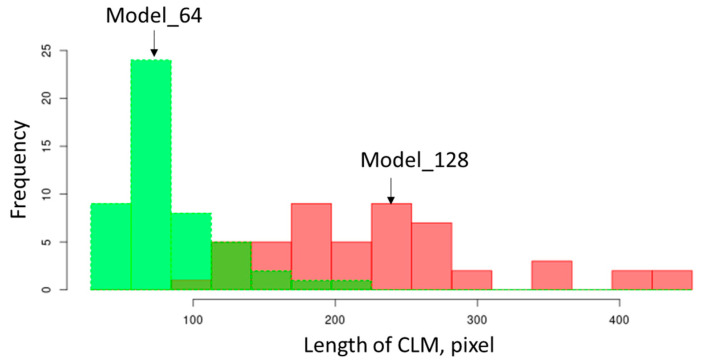
Length of CLM observed on YZ plane in model_64 and model_128 section images.

**Figure 10 jimaging-09-00090-f010:**
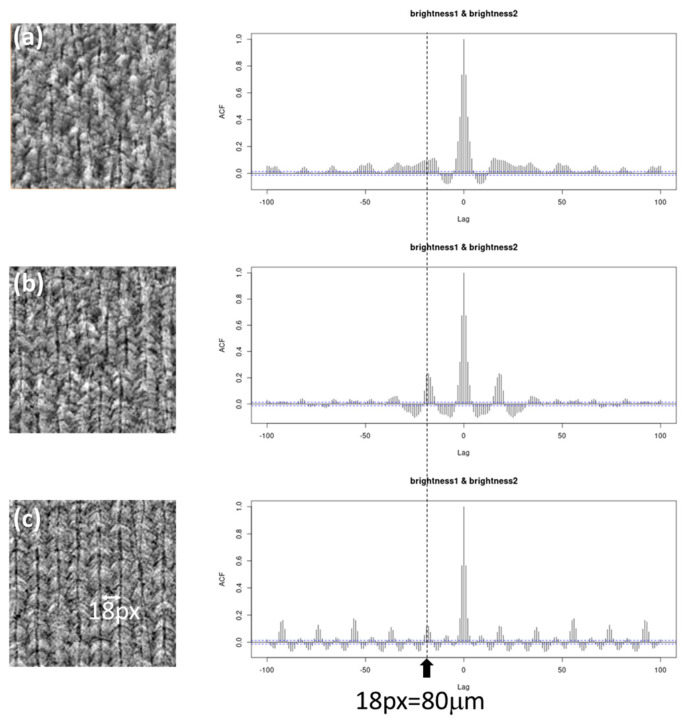
ACF analysis for the YZ section generated by (**a**) model_64 and (**b**) model_128 SliceGAN and (**c**) an original image.

**Figure 11 jimaging-09-00090-f011:**
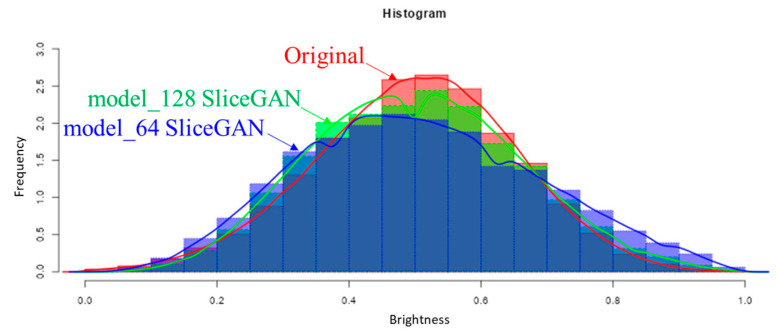
Comparison of brightness distribution between SliceGAN image and original image.

**Table 1 jimaging-09-00090-t001:** Architecture of model_64 SliceGAN.

Generator			
Layer	Function	Kernel, Stride, Padding	Output Shape
Input	-	-	64 × 4 × 4 × 4
1	ConvTranspose3d	4, 2, 2	512 × 6 × 6 × 6
	BatchNorm3d	-	
	ReLU	-	
2	ConvTranspose3d	4, 2, 2	256 × 10 × 10 × 10
	BatchNorm3d	-	
	ReLU	-	
3	ConvTranspose3d	4, 2, 2	
	BatchNorm3d	-	128 × 18 × 18 × 18
	ReLU	-	
4	ConvTranspose3d	4, 2, 2	64 × 34 × 34 × 34
	BatchNorm3d	-	
	ReLU	-	
5	ConvTranspose3d	4, 2, 3	1 × 64 × 64 × 64
	tanh	-	
**Critic**			
Layer	Function	Kernel, Stride, Padding	Output Shape
Input	-	-	1×64×64
1	Conv2d	4, 2, 1	64×32×32
	ReLU	-	
2	Conv2d	4, 2, 1	128×16×16
	ReLU	-	
3	Conv2d	4, 2, 1	256×8×8
	ReLU	-	
4	Conv2d	4, 2, 1	512×4×4
	ReLU	-	
5	Conv2d	4, 2, 0	1×1×1

**Table 2 jimaging-09-00090-t002:** Architecture of model_128 SliceGAN.

Generator			
Layer	Function	Kernel, Stride, Padding	Output Shape
Input	-	-	64 × 4 × 4 × 4
1	ConvTranspose3d	4, 2, 2	512 × 6 × 6 × 6
	BatchNorm3d	-	
	ReLU	-	
2	ConvTranspose3d	4, 2, 2	256 × 10 × 10 × 10
	BatchNorm3d	-	
	ReLU	-	
3	ConvTranspose3d	4, 2, 2	
	BatchNorm3d	-	128 × 18 × 18 × 18
	ReLU	-	
4	ConvTranspose3d	4, 2, 2	64 × 34 × 34 × 34
	BatchNorm3d	-	
	ReLU	-	
5	ConvTranspose3d	4, 2, 2	32 × 66 × 66 × 66
	BatchNorm3d	-	
	ReLU	-	
6	ConvTranspose3d	4, 2, 3	1 × 128 × 128 × 128
	tanh	-	
**Critic**			
Layer	Function	Kernel, Stride, Padding	Output Shape
Input	-	-	1 × 128 × 128
1	Conv2d	4, 2, 1	32 × 66 × 66
	ReLU	-	
2	Conv2d	4, 2, 1	64 × 32 × 32
	ReLU	-	
3	Conv2d	4, 2, 1	128 × 16 × 16
	ReLU	-	
4	Conv2d	4, 2, 1	256 × 8 × 8
	ReLU	-	
5	Conv2d	4, 2, 1	512 × 4 × 4
	ReLU	-	
6	Conv2d	4, 2, 0	1 × 1 × 1

**Table 3 jimaging-09-00090-t003:** Brightness distribution analysis.

	Mean	Standard Deviation
Original image (YZ section)	0.5001	0.1521
Model_64 SliceGAN image (YZ section)	0.5045 ± 0.0067	0.1789 ± 0.047
Model_128 SliceGAN image (YZ section)	0.5007 ± 0.0082	0.1552 ± 0.067

## Data Availability

The code that supports the findings of this study is available at https://nuss.nagoya-u.ac.jp/s/yi8XDfMjnTCLTmW (accessed on 25 April 2023).

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
