# Peer review of "Big-Volume SliceGAN for Improving a Synthetic 3D Microstructure Image of Additive-Manufactured TYPE 316L Steel"

_2313-433X, 2023, doi:10.3390/jimaging9050090_

Round 1

Reviewer 1 Report

This is an interesting manuscript in which  the SliceGAN architecture has been slightly adapted for generating non-binary 3D images of CLM from 2D microscopy images. The authors compare the results achieved by their architecture to those obtained by the original SliceGAN architecture.

The results seem to be interesting, however, the manuscript lacks precision, is vaguely formulated at many occasions and requires major revisions.

Here are just some examples:

Introduction: “Visualization” might be the wrong phrasing. It should be something like “reconstruction”. Another phrase could be “stochastic modeling”, since SliceGAN can be considered to be a stochastic geometry model.

Line 38: the latent variables are not multiplied by the number of channels. Be more precise. There are multiple occasions of this phrasing.

Line 47: “[…] are more grayscale than binarized images,”. This wording does not make sense. Binary images are not grayscale images, so how is the “degree of being a grayscale” image defined to make such a statement?

Lines 49-50: What does this sentence mean? Be more precise. What is an average feature of a microstructure?

Line 56-57. This is wrong. In [4] already some evaluation methods have been proposed. Moreover, there are numerous microstructure descriptors which have been considered in the literature to evaluate the discrepancy between a virtual microstructure generator and experimentally acquired data. See, for example, references provided in Chiu et al. (2013).

Sung Nok Chiu, Dietrich Stoyan, Wilfrid S. Kendall, Joseph Mecke. Stochastic Geometry and Its Applications, J. Wiley & Sons, 2013.

Line 68 “Five layers of transpose convolution are required to generate a […]“. This is not a requirement. There are other possibilities to transform the tensors into the desired shape.

Figure 2: This can be represented more efficiently. Just define a box for one discriminator training step. Then repeat this box 5 times (then you do not have to repeat the directional training steps).

Section 2: How are the batch-sizes to be interpreted? With a batch-size of 8 for the generator you generate 8 3D images?? Then 32 slices are cropped for the discriminator from the generated and real 2D images??

Furthermore, provide more details on the training: Which optimizer was used, which training rates were used, etc.

Line 114: This formula is wrong. What’s “t”? t and i should probably be the same. Moreover, the authors never mention that x denotes their images. Are those 2D or 3D images. Are the positions mentioned 1D, 2D, or 3D coordinates?

According to their plots, they seem to be 1D coordinates, but how were these values computed from images which might have 2D or 3D coordinates? Additionally, the definition of the autocorrelation using an integral is not suited for discretized signals.

Lines 115-118: Provide more details on the brightness distribution analysis.

Section 4: What type of image acquisition technique was used? What were the acquisition parameters? If the material was only measured in x-direction, how have three perpendicular images been acquired?

Line 123: “the laser was scanned”. The laser is not scanned. A material could be scanned by a laser.

Line 136-138: This “seemingly 3D image” seems to be just a visualization. How exactly is this type of visualization used for a quantitative comparison? It is mentioned here, but it seems to be illusive how this type of visualization helps with the quantitative comparison.

Section 5: It is unclear, how exactly the additional EBSD measurements are relevant for this work.

Lines 152-164: This entire paragraph is too unprecise and vague:

-   Lines 152-153: Figure 6 does not show 3D images. It shows 2D sections of 3D images

-   Lines 153-154: What does this mean? Which “surface” has been generated? How is the reference to Figures 7 and 8 to be interpreted? The sentence suggests that the images have been generated according to some procedure visualized in Figures 7 and 8, but this seems not to be the case.

-   What exactly is meant by “almost seems straight”? Be more precise. Furthermore, give more details on the discontinuities. Additionally, the visualization provided in Figures 6 and 7 do not clearly show these effects. They should be adapted accordingly

-   Line 180: What does “terminated” mean exactly in this context?

Figure 9: This visualization is not well suited for a comparison. Plot two curves in the same figure, for better comparability.

Author Response

Response to the reviewer’s comments

We would like to thank the reviewer for careful and thorough reading of our manuscript and for valuable comments and suggestions. Our specific responses to each comment are as follows (the reviewer’s comments are in italics).

To reviewer 1

General Comments: This is an interesting manuscript in which the SliceGAN architecture has been slightly adapted for generating non-binary 3D images of CLM from 2D microscopy images. The authors compare the results achieved by their architecture to those obtained by the original SliceGAN architecture.

Reply: We are grateful to the reviewer for his/her encouraging comments. The comments and suggestions have been carefully considered and the corresponding revisions have been made in the following as suggested by the reviewer.

Comment 1: Introduction: “Visualization” might be the wrong phrasing. It should be something like “reconstruction”. Another phrase could be “stochastic modeling”, since SliceGAN can be considered to be a stochastic geometry model.

Reply: Thank you for your suggestion. We have changed the word “visualization” to “reconstruction”.

Comment 2: Line 38: the latent variables are not multiplied by the number of channels. Be more precise. There are multiple occasions of this phrasing.

Reply: We completely agree with reviewer’s comment. We changed the expression “4 × 4 × 4 latent variables multiplied by 64 channels” to “64 sets of latent variables in the format of 4 × 4 × 4 (voxel)”.

Comment 3: Line 47: “[…] are more grayscale than binarized images,”. This wording does not make sense. Binary images are not grayscale images, so how is the “degree of being a grayscale” image defined to make such a statement?

Reply: Thank you for pointing this out. We have changed the corresponding sentence as follows. “Since actual microstructural photographs are grayscale images rather than binary images, it would be more useful in material research to improve the quality of a Slice-GAN-generated image using grayscale 2D images.”  

Comment 4: Lines 49-50: What does this sentence mean? Be more precise. What is an average feature of a microstructure?

Reply: Thank you for reviewer’s valuable comment. We have changed the corresponding sentence to make the meaning clear. “In particular, generating a large representative volume while maintaining high resolution is beneficial for reproducing a representative microstructure.”

Comment 5: Line 56-57. This is wrong. In [4] already some evaluation methods have been proposed. Moreover, there are numerous microstructure descriptors which have been considered in the literature to evaluate the discrepancy between a virtual microstructure generator and experimentally acquired data. See, for example, references provided in Chiu et al. (2013).

Sung Nok Chiu, Dietrich Stoyan, Wilfrid S. Kendall, Joseph Mecke. Stochastic Geometry and Its Applications, J. Wiley & Sons, 2013.

Reply: Thank you for your valuable comment and letting us to know a reference. Following reviewer’s suggestion, we added the following sentence in a text. “However, there is currently little established method for evaluating grayscale 3D images. Kench and Cooper [4] suggested a possible approach to measure the similarity between an experimentally obtained 3D image and a synthetic 3D image generated by SliceGAN. They used tomographic 3D data collected from a Li-ion MMC cathode sample and trained SliceGAN with a random subset of 2D sections of the 3D image. They then evaluated the similarity between the real and synthetic 3D images based on volume fraction, relative surface area, and relative diffusivity. This assessment demonstrated that SliceGAN can potentially generate a 3D image that is similar to the ground truth. However, their evaluation only used binary 3D images, and the quality of synthetic 3D images with grayscale sections is still under debate.”

Comment 6: Line 68 “Five layers of transpose convolution are required to generate a […]“. This is not a requirement. There are other possibilities to transform the tensors into the desired shape.

Reply: Many thanks to the reviewer’s good suggestion. We have changed the corresponding sentence as follows. “The original model uses five layers of transpose convolution to generate a 1 channel × 64 × 64 × 64 voxel 3D fake image from 64 sets of latent variables in the format of 4 × 4 × 4.”

Comment 7: Figure 2: This can be represented more efficiently. Just define a box for one discriminator training step. Then repeat this box 5 times (then you do not have to repeat the directional training steps).

Reply: We appreciate reviewer’s good suggestion. We have modified Figure 2 to make it simpler.

Comment 8: Section 2: How are the batch-sizes to be interpreted? With a batch-size of 8 for the generator you generate 8 3D images?? Then 32 slices are cropped for the discriminator from the generated and real 2D images??

Reply: Indeed, the explanation here was not clear. We have modified the sentence as follows. “Both models use thirty-two subdivided datasets (batch size) for generator training, per-formed only once. Meanwhile, eight subdivided datasets are used for critic training, which is repeated five times in one iteration. The optimizer used is Adam, as in the original SliceGAN model [4].”

Comment 9: Furthermore, provide more details on the training: Which optimizer was used, which training rates were used, etc.

Reply: We have added the following explanation. “The optimizer used is Adam, as in the original SliceGAN model [4].”

Comment 10: Line 114: This formula is wrong. What’s “t”? t and i should probably be the same. Moreover, the authors never mention that x denotes their images. Are those 2D or 3D images. Are the positions mentioned 1D, 2D, or 3D coordinates?

Reply: We completely agree with reviewer’s comment on this issue. We have corrected the sentence as follows. “The ACF expresses the relationship between the overlap and staggered distance (called lag) of these images. The conventional correlation (rxy) is somewhat modified for ACF because X and Y are the same, except for the lag, and is given by the following equation:

,

The equation for conventional correlation uses sample means ( and ) and sample standard deviations ( and ) and is modified for ACF as  and  are the same, except for the lag.”

Comment 11: According to their plots, they seem to be 1D coordinates, but how were these values computed from images which might have 2D or 3D coordinates? Additionally, the definition of the autocorrelation using an integral is not suited for discretized signals.

Reply: We added the following sentence to explain the method to get Figure 8.

 “To compute ACF, two images are shifted along the direction perpendicular to CLM.”

Comment 12: Lines 115-118: Provide more details on the brightness distribution analysis.

Reply: We appreciate reviewer’s good suggestion. We added some sentences on this issue. “Therefore, in this study, we analyzed the density distribution of brightness values (Figure 11). The density distribution of brightness values for the three images (original, model_64, and model_128 images) is considered to be a Gaussian distribution with one peak. The density distribution of the Model_128 SliceGAN image is similar to that of the original image, while that of the Model_64 SliceGAN image appears to be wider. To examine the brightness distribution more quantitatively, we obtained the mean value and standard deviation of brightness for the three images, and the results are presented in Table 3. Both the mean value and standard deviation are quite similar between the original image and the Model_128 SliceGAN image. The mean and standard deviation of the brightness distribution of the Model_64 SliceGAN image are slightly larger than those of the original image. In the future, a more quantitative analysis of gray-scale 2D/3D images is desired.”  

Comment 13: Section 4: What type of image acquisition technique was used? What were the acquisition parameters? If the material was only measured in x-direction, how have three perpendicular images been acquired?

Reply: We examined the similarity between a ground truth 2D image and 2D section image sliced from synthetic 3D SliceGAN image by auto-correlation function (ACF) analysis and brightness distribution analysis. In addition, we measured the length of a characteristic microstructure of both a ground truth 2D image and 2D section image sliced from synthetic 3D SliceGAN image. These explanations were added in the text. 

Comment 14: Line 123: “the laser was scanned”. The laser is not scanned. A material could be scanned by a laser.

Reply: We agree reviewer’s comment. We have corrected the corresponding sentence. “the powder was scanned by a laser only in the X direction using Scan strategy X [15] (Figure 3).”

Comment 15: Line 136-138: This “seemingly 3D image” seems to be just a visualization. How exactly is this type of visualization used for a quantitative comparison? It is mentioned here, but it seems to be illusive how this type of visualization helps with the quantitative comparison.

Reply: An explanation was added as follows. “This pseudo-3D image is helpful for understanding the relationship between the three sections and will be compared with a 3D image generated by SliceGAN later on.”

Comment 16: Section 5: It is unclear, how exactly the additional EBSD measurements are relevant for this work.

Yes, we also think that it is important to reconstruct a color image in 3D. We added a following explanation in the text. “One of RGB values is a kind of grayscale brightness value. Therefore, a high-quality synthetic color 3D image generated by SliceGAN seems to be very attractive. Kench and Cooper [4] demonstrated that SliceGAN could generate a 3D image from 2D color images as well. However, the quality of the color 3D SliceGAN image seems to be not good compared with a binary image. To realize to improve the quality of the color 3D SliceGAN image, “Big-volume SliceGAN” is likely useful. But it requires much more GPU memory than 48 GB and it is beyond the scope of this study.”      

Comment 17: Lines 152-164: This entire paragraph is too unprecise and vague:

We modified our explanation in this part as follows. “Model_128 produced a higher quality image compared to model_64, particularly for CLM, which appears more continuous in the model_128 image than in the model_64 image. Although a 128 × 128 (pixel) image was used as input for the training of SliceGAN, it is surprising that some CLMs appear continuous in the 512 × 512 (pixel) section of the model_128 image. To quantitatively evaluate the continuity of CLM observed in SliceGAN images, the length of fifty CLMs was measured for each image of model_64 and model_128 on the YZ plane. The potential maximum length of CLM is 512 (pixel) due to the image size. Figure 9 shows that the CLM in the model_128 image is much longer than that in the model_64 image. The average lengths of CLM in the image and in model_64 images are 233 and 83 (pixel), respectively.”

Comment 18: -   Lines 152-153: Figure 6 does not show 3D images. It shows 2D sections of 3D images

Reply: Following the reviewer’s suggestion, we changed the corresponding sentence. “The sections on the YZ, XY, and XZ planes of the generated 3D images are shown in Figures 7 and 8.”

Comment 19: Lines 153-154: What does this mean? Which “surface” has been generated? How is the reference to Figures 7 and 8 to be interpreted? The sentence suggests that the images have been generated according to some procedure visualized in Figures 7 and 8, but this seems not to be the case.

Reply: We have added an explanation as described above.

Comment 20: What exactly is meant by “almost seems straight”? Be more precise. Furthermore, give more details on the discontinuities. Additionally, the visualization provided in Figures 6 and 7 do not clearly show these effects. They should be adapted accordingly

Comment 21: Line 180: What does “terminated” mean exactly in this context?

Reply: Thank you for reviewer’s valuable suggestions. CLM length observed on YZ plane in model_64 and model_128 section images was additionally measured, which result was shown in Figure 9. Also, the following explanation was added in a sentence. “To quantitatively evaluate the continuity of CLM observed in SliceGAN images, the length of fifty CLMs was measured for each image of model_64 and model_128 on the YZ plane. The potential maximum length of CLM is 512 (pixel) due to the image size. Figure 9 shows that the CLM in the model_128 image is much longer than that in the model_64 image. The average lengths of CLM in the image and in model_64 images are 233 and 83 (pixel), respectively.”

Comment 22: Figure 9: This visualization is not well suited for a comparison. Plot two curves in the same figure, for better comparability.

Reply: Following reviewer’s suggestion, the corresponding figure was modified.

Reviewer 2 Report

In this manuscript, a modified SliceGAN architecture is proposed to obtain a high-quality synthetic three-dimensional (3D) microstructure image of an additive-manufactured TYPE 316L steel. The specific material is selected because it has a strongly anisotropic microstructure that requires 3D visualization. It is shown that doubling the training image size while maintaining a resolution permits to obtain a more realistic synthetic 3D image.

The manuscript is interesting and well written. It could be further improved after revision according to the following comments:

* In Section 2, more details (and the respective equations) should be provided regarding the updating of the weight coefficients of the 3D generator and the computational cost of the training.

* The figures should be renumbered after Figure 5. Moreover, current Figure 9 and ref. [16] are not cited-discussed in the text. This should be fixed.

* In the Introduction, “Kench [4]” should be corrected to “Kench and Cooper [4]”.

* In p. 1, line 32 and p. 5, line 128, it would be better to write “perpendicular planes” or “perpendicular sections” instead of “perpendicular faces”.

* p. 3, line 83: I think that “4 x 4 latent variables” should be changed to “4 x 4 x 4 latent variables”.

Author Response

Response to the reviewer’s comments

We would like to thank the reviewer for careful and thorough reading of our manuscript and for valuable comments and suggestions. Our specific responses to each comment are as follows (the reviewer’s comments are in italics).

To reviewer 2

General Comments: In this manuscript, a modified SliceGAN architecture is proposed to obtain a high-quality synthetic three-dimensional (3D) microstructure image of an additive-manufactured TYPE 316L steel. The specific material is selected because it has a strongly anisotropic microstructure that requires 3D visualization. It is shown that doubling the training image size while maintaining a resolution permits to obtain a more realistic synthetic 3D image.

The manuscript is interesting and well written. It could be further improved after revision according to the following comments:

Reply: Many thanks to the reviewer for the positive comments. The comments have been carefully considered and the corresponding revision in the following has been made as suggested by the reviewer.

Comment 1: In Section 2, more details (and the respective equations) should be provided regarding the updating of the weight coefficients of the 3D generator and the computational cost of the training.

Reply: We appreciate reviewer’s valuable comment. The following explanation was added. “After this repeated training was completed, 64 sets of latent variables in the format of 18 × 18 × 18 and 10 × 10 × 10 (voxel) were input to the 3D-generators of model_64 and model_128, respectively, to obtain a larger 3D synthetic image with a 512 × 512 × 512 (voxel) multiplied by 1 channel. In both models, the thirty-two subdivided datasets (batch size) were used for training the generator, which was performed only once. Meanwhile, eight subdivided datasets were used for training the critic, which was repeated five time in one iteration. The used optimizer was Adam according to the original SliceGAN model [4].”

Comment 2: The figures should be renumbered after Figure 5. Moreover, current Figure 9 and ref. [16] are not cited-discussed in the text. This should be fixed.

Reply: Thank you for pointing out our mistake in numbering figures. We have corrected it and ref.[16] was cited in the text as follows. “There are two main types of image generation algorithms: one is an adversarial generation network (GAN) [5] and the other is a variational autoencoder [16].”

Comment 3: In the Introduction, “Kench [4]” should be corrected to “Kench and Cooper [4]”.

Reply: We corrected the reference.

Comment 4: In p. 1, line 32 and p. 5, line 128, it would be better to write “perpendicular planes” or “perpendicular sections” instead of “perpendicular faces”.

Reply: Following reviewer’s suggestion, we change the word “faces” to “planes”.

Comment 5: p. 3, line 83: I think that “4 x 4 latent variables” should be changed to “4 x 4 x 4 latent variables”.

Reply: Thank you for your finding our careless mistake. We have corrected the text.

Round 2

Reviewer 1 Report

The authors took all comments into consideration when revising
their paper. It is now substantially improved and can be accepted
for publishing.

Reviewer 2 Report

The revised manuscript has been improved according to the reviewer's comments and is recommended for publication in its present form.